## PERSPECTIVES

# Developing in a polluted atmosphere: A link between long-term exposure to elevated atmospheric CO$_2$ and hyperactivity

Helen Stolp 

*Department of Comparative Biomedical Sciences, Royal Veterinary College, London, UK*

Email: hstolp@rvc.ac.uk

Edited by: Laura Bennet & Justin Dean

Linked articles: This Perspectives article highlights an article by Wyrwoll *et al.* To read this paper, visit https://doi.org/10.1113/JP282179.

The peer review history is available in the Supporting Information section of this article (https://doi.org/10.1113/JP282577#support-information-section).

The World Health Organisation (WHO) estimates that 99% of people worldwide breathe polluted air on a day-to-day basis (i.e. air with pollutant levels higher than WHO guidelines) and that ambient air pollution causes more than 4 million premature deaths per year. These numbers reflect the effect of particulate matter on cardiovascular and respiratory disease (Cohen *et al.* 2017) and probably underestimate the importance of air pollution on health because the role of atmospheric carbon dioxide (CO$_2$) has not been considered thus far. Furthermore, air pollution has neurotoxic effects that are increasingly recognised and elevated atmospheric CO$_2$ has the potential to cause renal and bone disease, expanding the latent health impact of air pollution globally.

In this issue of *The Journal of Physiology*, Wyrwoll *et al.* (2022) address this timely question, exploring both systemic and neurological outcomes of a life lived in a high CO$_2$ environment. Comparing life-time exposure to current atmospheric CO$_2$ (460 ppm) with the higher CO$_2$ level of 890 ppm, a level predicted to occur by the year 2100, Wyrwoll *et al.* (2022) showed that mice exposed to elevated CO$_2$ during gestation had a higher birth weight than their control counterparts. Abnormal birth weight can increase risk of later morbidity, although, in this case, weight normalised over the postnatal period and

developmental milestones, such as righting reflex or eye opening, were not delayed. Dimorphism occurred between males and females at adulthood, with high CO$_2$ exposed females having lower body weight compared to age-matched controls.

It is well known that substantial acute increases in inspired, or decreases in expired, CO$_2$ result in respiratory acidosis, where the pH of the blood falls below the normal physiological range. Short-term compensations include increased respiration or altered renal excretion of hydrogen and bicarbonate ions. Longer-term, excess hydrogen can be buffered by bicarbonate released from bone. This bicarbonate release may lead to osteoporosis, whereas the accompanying release of calcium and other ions can contribute to renal calcification. It has been unclear until now whether increased atmospheric CO$_2$ could drive these pathologies. This question is pertinent given that the world is experiencing unprecedented increases in atmospheric CO$_2$.

Surprisingly, no pathological changes were observed in kidney structure at adulthood following exposure to the high CO$_2$ environment for 3 months (i.e. from gestation to young adulthood) or using measures of renal function (Wyrwoll *et al.* 2022). Nor were differences in bone structural density identified, as would be expected if substantial physiological compensation was occurring. This may reflect the relatively short experimental period and the fact that blood pH was only borderline acidotic in this model, although it was sufficient to change the structure and function of the lungs, as detailed in a companion study (Larcombe *et al.* 2021).

Particulate matter 2.5 (PM2.5) from air pollution has now been added to the list of environmental pollutants with neurotoxic capacity (e.g. pesticides, heavy metals). A plethora of recent studies show that PM2.5 can produce inflammation and oxidative stress throughout the brain, linked to increased risk of neurodevelopmental disorders and neurodegenerative disease (Costa *et al.* 2020). Acute increases in CO$_2$, as may arise in a poorly ventilated indoor setting, can reduce cognitive function, particularly decision making and executive function (Karnauskas *et al.* 2020). This reduction is probably a result of altered

neuronal excitability, a phenomenon that could affect brain development, where neural activity is essential for co-ordinating normal developmental events. The study by Wyrwoll *et al.* (2022) makes the case that global rising CO$_2$ levels could confer a risk similar to other components of air pollution.

High life-time CO$_2$ exposure produced significant differences in dopamine receptors in the brain of adult offspring (Wyrwoll *et al.* 2022). Increased expression of the dopamine receptor D1 (DRD1) was found in males, whereas DRD2 expression decreased in females. Increased DRD1 activity has been associated with hyperactivity, a behaviour observed in both male and female mice in the study by Wyrwoll *et al.* (2022). Plasma corticosterone levels also increased with CO$_2$ exposure, possibly contributing to the hyperactivity. Expression of the gene for 11$\beta$-hydroxysteroid dehydrogenase type 1 (Hsd11b1), an enzyme that converts cortisone to cortisol, was increased in the hippocampus, further supporting this possibility. Chronic elevation of brain glucocorticoids associates with age-related neurodegeneration and cognitive impairment, and could harm the developing brain.

Further exploration is needed to determine how elevated CO$_2$ may affect pregnancy, ageing and the mechanisms underlying behavioural change. It is unclear from the study by Wyrwoll *et al.* (2022) whether the reported hyperactivity is a result of changes in brain structure and function. Elucidating both behaviours and their neurological basis in this paradigm will not be trivial. The interpretation of behaviour in mice is always complicated and, in the study by Wyrwoll *et al.* (2022), CO$_2$ exposed mice appeared to interact abnormally with some behavioural testing paradigms in a manner not reflected in the test metrics.

One limitation of the study by Wyrwoll *et al.* (2022) was the choice to use 460 ppm CO$_2$ as the control condition. Average CO$_2$ in the recent past was lower, ~370 ppm at the turn of the century, and, although 460 ppm is a feasible estimate of current ambient CO$_2$ within an urban environment (average atmospheric CO$_2$ of 420 ppm plus local urban production), this choice may reflect an already

The Journal of Physiology

increased baseline, potentially normalising pathological adaptations occurring within the population and downplaying the consequences of rising atmospheric $CO_2$. The study by Wyrwoll *et al.* (2022) represents the tip of the iceberg of what we know about the effect of prolonged exposure to elevated atmospheric $CO_2$. The importance of this will inevitably become clearer over the coming years.

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

## Additional information

### Competing interests

No competing interests declared.

### Author contributions

Sole author.

### Funding

No funding was received.

### Keywords

acid-base compensatory physiology, carbon dioxide, corticosterone, neurotoxicity

## Supporting information

Additional supporting information can be found online in the Supporting Information section at the end of the HTML view of the article. Supporting information files available:

**Peer Review History**

