## [Peer Review History · The Journal of Physiology]

Developing in a polluted atmosphere: A link between long-term exposure to elevated atmospheric CO₂ and hyperactivity

Helen B Stolp

DOI: 10.1113/JP282577

Corresponding author(s): Helen Stolp (hstolp@rvc.ac.uk)

Review Timeline:

Submission Date:	17-Nov-2021
Editorial Decision:	30-Nov-2021
Revision Received:	07-Dec-2021
Editorial Decision:	16-Dec-2021
Revision Received:	24-Dec-2021
Accepted:	05-Jan-2022

Senior Editor: Laura Bennet

Reviewing Editor: Justin Dean

Transaction Report:

Dear Dr Stolp,

Re: JP-P-2021-282577 "Developing in a bad atmosphere: A link between long-term exposure to elevated atmospheric CO2 and hyperactivity" by Helen B Stolp

Thank you for submitting your invited Perspectives article to The Journal of Physiology. It has been assessed by a Reviewing Editor and the author of the focus paper.

Minor alterations have been requested.

The reports are copied at the end of this email. Please address all of the points and incorporate all requested revisions.

NEW POLICY: In order to improve the transparency of its peer review process The Journal of Physiology publishes online as supporting information the peer review history of all articles accepted for publication. Readers will have access to decision letters, including all Editors' comments and referee reports, for each version of the manuscript and any author responses to peer review comments. Referees can decide whether or not they wish to be named on the peer review history document.

I hope you will find the comments helpful and have no difficulty in revising your article within 7 days.

To submit the revised version use the links in Author Tasks Link Not Available.

Please ensure that the article is a Word File with no more than 5 references, including the focus paper.

Thank you for your contribution to the Journal.

Yours sincerely,

Professor Laura Bennet
Senior Editor
The Journal of Physiology
<https://jp.msubmit.net>
<http://jp.physoc.org>
The Physiological Society
Hodgkin Huxley House
30 Farringdon Lane
London, EC1R 3AW
UK
<http://www.physoc.org>
<http://journals.physoc.org>

EDITOR COMMENTS

Reviewing Editor:

Please address the minor comments provided by the reviewer.

REFEREE COMMENTS:

Referee #1:

Thank you for the Perspective, it reflects the significance of the work well. A few minor comments below.

The use of "bad" in the title is unclear. Perhaps "polluted" would be a better alternative?

Paragraph 4 - maybe could also mention that the exposure duration was only a few months and that renal and osteological compensation may take longer to kick in. 3 months is "young-adult" in a mouse.

Second last paragraph - it is mentioned we said rodents can smell CO₂. C57 mice (used in this study) have an effectively nil olfactory receptor response at CO₂ < 1000 ppm (i.e. so it's unlikely the behavioural changes we saw are because of them smelling the CO₂).

Tuesday, 07 December 2021

Professor Laura Bennet
Editor, The Journal of Physiology

Dear Professor Bennet,

I would like to thank you for commissioning the manuscript “Developing in a bad atmosphere: A link between long-term exposure to elevated atmospheric CO₂ and hyperactivity”.

I would also like to thank the reviewers for their constructive suggestions for improvement to this Perspective. I have incorporated all the suggested changes, as outlined in detail in the Tracked Changes version of the document.

Specifically, I have:

1. changed the title to: “Developing in a polluted atmosphere: A link between long-term exposure to elevated atmospheric CO₂ and hyperactivity”
2. add a reference in the 4th paragraph to the 3-month treatment period and the significance of this on the findings of the study by Wyrwoll et al. (2021)
3. removed the comment about rodents smelling CO₂ from the second last paragraph, as it doesn’t add the to the point discussed and is addressed in detail by Wyrwoll et al. (2021).

I hope you now consider this work acceptable for publication in *The Journal of Physiology*. I look forward to hearing from you in due course.

Yours faithfully,

Dr Helen Stolp

Dear Dr Stolp,

Re: JP-P-2021-282577R1 "Developing in a polluted atmosphere: A link between long-term exposure to elevated atmospheric CO2 and hyperactivity" by Helen B Stolp

Thank you for submitting your invited Perspectives article to The Journal of Physiology. It has been assessed by a Reviewing Editor and the author of the focus paper.

Minor alterations have been requested.

The reports are copied at the end of this email. Please address all of the points and incorporate all requested revisions.

NEW POLICY: In order to improve the transparency of its peer review process The Journal of Physiology publishes online as supporting information the peer review history of all articles accepted for publication. Readers will have access to decision letters, including all Editors' comments and referee reports, for each version of the manuscript and any author responses to peer review comments. Referees can decide whether or not they wish to be named on the peer review history document.

I hope you will find the comments helpful and have no difficulty in revising your article within 7 days.

To submit the revised version use the links in Author Tasks Link Not Available.

Please ensure that the article is a Word File with no more than 5 references, including the focus paper.

Thank you for your contribution to the Journal.

Yours sincerely,

Professor Laura Bennet
Senior Editor
The Journal of Physiology
<https://jp.msubmit.net>
<http://jp.physoc.org>
The Physiological Society
Hodgkin Huxley House
30 Farringdon Lane
London, EC1R 3AW
UK
<http://www.physoc.org>
<http://journals.physoc.org>

EDITOR COMMENTS

Reviewing Editor:

Thank you for your changes, the author is happy with changes but I suggest some minor changes for grammar:

Paragraph 1, Line 5: "...of air pollution on health as the role of atmospheric..." replace with "...of air pollution on health, as the role of atmospheric..."

Paragraph 4 Line 5: "...though sufficient to changes the structure and function..." replace with "...though sufficient to change the structure and function..."

Paragraph 5 Line 2: "...pollutants with neurotoxic capacity (e.g. pesticides, heavy metals)." replace with "...pollutants with neurotoxic capacity (e.g., pesticides, heavy metals)."

In addition, please present your References in journal style (<https://jp.msubmit.net/cgi-bin/main.plex?>

form_type=display_requirements#references). The references should be given in full in the text (e.g. 'Smith et al., 2021') and then listed alphabetically in the References section. They should not be numbered, as they are currently.

REFEREE COMMENTS:

Referee #1:

I am happy with the revisions.

1st Confidential Review

07-Dec-2021

Dr Helen Stolp
Lecturer in Pharmacology
Comparative Biomedical Sciences
Royal Veterinary College
London, NW1 0TU

Friday, 24 December 2021

Professor Laura Bennet
Editor, The Journal of Physiology

Dear Professor Bennet,

I would like to thank you for commissioning the manuscript “Developing in a bad atmosphere: A link between long-term exposure to elevated atmospheric CO₂ and hyperactivity”.

I have incorporated all the suggested changes. I hope you now consider this work acceptable for publication in *The Journal of Physiology*. I look forward to hearing from you in due course.

Yours faithfully,

Dr Helen Stolp

Dear Dr Stolp,

Re: JP-P-2021-282577R2 "Developing in a polluted atmosphere: A link between long-term exposure to elevated atmospheric CO₂ and hyperactivity" by Helen B Stolp

I am pleased to tell you that your invited Perspective article has been accepted for publication in The Journal of Physiology.

NEW POLICY: In order to improve the transparency of its peer review process The Journal of Physiology publishes online as supporting information the peer review history of all articles accepted for publication. Readers will have access to decision letters, including all Editors' comments and referee reports, for each version of the manuscript and any author responses to peer review comments. Referees can decide whether or not they wish to be named on the peer review history document.

The last Word version of the paper submitted will be used by the Production Editors to prepare your proof. When this is ready you will receive an email containing a link to Wiley's Online Proofing System. The proof should be checked and corrected as quickly as possible.

All queries at proof stage should be sent to tjp@wiley.com

Thank you very much for your contribution to The Journal of Physiology.

Yours sincerely,

Professor Laura Bennet
Senior Editor
The Journal of Physiology
<https://jp.msubmit.net>
<http://jp.physoc.org>
The Physiological Society
Hodgkin Huxley House
30 Farringdon Lane
London, EC1R 3AW
UK
<http://www.physoc.org>
<http://journals.physoc.org>